# DTL-IceNet: A Dual-Task Learning Architecture with Multi-

# 2 Scale Fusion Mechanisms for Enhanced Ice Detection on

# **3 Transmission Lines**

- Yufei Fu<sup>1</sup>, Yang Cheng<sup>1</sup>, SongYuan Cao<sup>2</sup>, Ling Tan<sup>3</sup>, Jiaxin He<sup>3</sup>, Mengya Wang<sup>3</sup>, Wenjie
- Zhang<sup>3</sup>
- <sup>1</sup>Electric Power Research Institute, State Grid Anhui Electric Power Company Ltd., Hefei, China.
- <sup>2</sup>State Grid Anhui Electric Power Company Ltd., Hefei, China.
- <sup>3</sup>School of Computer Science, Nanjing University of Information Science and Technology, Nanjing, China.
- Correspondence to: Wenjie Zhang (zhangwenjie@nuist.edu.cn)
- Abstract. Icing on transmission lines can significantly impact the stable operation of the power system. Deep
- learning-based ice image recognition is effective but remains vulnerable to background interference and noise,
- degrading accuracy. Moreover, when detecting ice thickness, the 2D nature of ice images introduces spatial
- limitations in representing the 3D ice state, which can lead to detection errors caused by a single viewpoint. To
- tackle the aforementioned challenges, this paper proposes DTL-IceNet (Dual-Task Learning Ice Detection Network),
- a transmission line icing detection network based on a dual-task learning framework, designed to accurately identify
- both the type and thickness of ice on overhead transmission lines. DTL-IceNet incorporates a multi-branch
- structured ice coating recognition module, ResSepNet (Residual & Depth-Separable Convolution Network), which
- segments the background and conductor areas to mitigate the influence of background noise. Additionally, a
- semantic segmentation module, MOMSA-SegNet (MobileOne & Multi-Scale Attention Segmentation Network) is
- designed to segment the ice-covered areas in both the main and side views of the image. The multi-scale attention
- mechanism is employed to extract spatial features from the raw icing image. When calculating ice thickness, the
- multi-scale fusion and correction optimization are adopted to enhance the algorithm. Experimental results show that
- compared with other models, the proposed method achieves an improvement of 4.17 % in icing type identification
- accuracy and a MAPE of 11.82 % in icing thickness detection. The application of this approach is crucial for
- reducing the hazards caused by ice coating on transmission lines and improving the stability of the power grid.

#### 1 Introduction

- Extreme weather can lead to ice accumulation on power lines, significantly increasing the risk of incidents such as
- conductor breakage or tower collapse, thereby threatening the stability of the power supply. Therefore, real-time
- monitoring of ice type, thickness, and other conditions on transmission lines is essential for ensuring the safe and
- stable operation of the power grid.
- Traditional ice detection methods primarily rely on physical sensors and manual inspections (Zhang et al., 2024).
- However, these methods often suffer from high costs, low real-time performance, and limited detection accuracy,
- making them insufficient for effective ice monitoring in complex environments. In recent years, with the rapid

https://doi.org/10.5194/egusphere-2025-3097 Preprint. Discussion started: 13 November 2025 © Author(s) 2025. CC BY 4.0 License.

advancement of deep learning and computer vision technologies, intelligent detection methods based on the YOLO model have increasingly become an effective approach. Chen et al. (2024) proposed a transmission line icing detection method based on YOLOv8. They utilized the ghost shuffle convolution to reduce model parameters and improve computational efficiency. Additionally, they incorporated the BiFormer attention mechanism and the Wise-IoUv3 loss function to enhance the model's accuracy in detecting ice-covered areas. Kong et al. (2024) integrated the GE attention module into YOLOv8 to enhance detection accuracy and replaced the concatenate structure in the original network with the BiFPN feature fusion module. This modification enables the detection of ice-covered areas on power transmission lines in complex backgrounds. Although the YOLO-based detection algorithm effectively locates ice-covered areas on transmission lines, it fails to detect and assess key information, such as ice contours and thickness. Building on this, Lu (2024) proposed the Canny-UNet model by enhancing YOLOv8 with EfficientViT (Liu et al., 2023), and integrating the Canny edge detection algorithm along with semantic segmentation technology, which further enabled accurate segmentation of ice contours. Similarly, He et al. (2023) applied the ProtoNet segmentation model to the detection results of the improved YOLOv5s, enabling the segmentation of ice-covered areas based on target detection. Similarly, He et al. (2023) utilized the GrabCut algorithm in conjunction with target detection to identify and segment transmission line insulators.

Although the aforementioned methods employ edge detection and semantic segmentation techniques to segment and detect the contours of ice-covered regions, the calculation of ice thickness primarily depends on edge detection algorithms. Wang et al (2023) proposed an image denoising algorithm based on adaptive switching median filter. Building upon this, an optimized Canny operator was employed to detect the edges of the ice-covered conductor's contour. The computed ice thickness was then compared with the results obtained from optical fiber detection, yielding an average error of just 4.10 %. Yang et al. (2023) proposed an ice monitoring method integrating image edge detection and normal detection. The approach first preprocesses micro-photographed images of transmission lines, applies algorithms such as eight-neighborhood tracking to detect edges and determine the longest side of the conductor, and designs an ice thickness detection method based on edge normal detection. Experimental results indicate that the relative error of real-time conductor ice thickness measurements using this method does not exceed 9 %. He et al. (2023) proposed a novel measurement method for thickness of uneven icing on transmission line in complex background. Their method involved image grizzling, median filter denoising, and maximum inter-class variance method to analyze the images. By integrating the result-domain characteristics of transmission line icing information and background noise, they extracted the re-icing transmission line. Finally, the vertical line approximation method was applied to determine the re-icing thickness. Such methods leverage edge detection techniques to enhance the extraction of ice cover information and initially estimate the corresponding ice thickness. However, they exhibit limited robustness to environmental interferences such as lighting variations and haze and fail to account for the three-dimensional spatial distribution of the conductors. Consequently, when encountering irregular ice formations, these methods may yield larger errors.

Accurately identifying the type of ice on transmission lines is crucial for improving ice detection capabilities. In the field of ice classification, some researchers analyze monitoring data to distinguish different ice types. Fan et al. (2018) analyzed the collision rate of water droplets on conductors with varying diameters and employed the standard

https://doi.org/10.5194/egusphere-2025-3097 Preprint. Discussion started: 13 November 2025 © Author(s) 2025. CC BY 4.0 License.

ice thickness normalization method to quantify the extent of conductor icing. Hao et al. (2023) analyzed multi-source data and applied the KNN algorithm to classify four distinct types of ice cover. Chen et al. (2024) proposed a method to monitor the status of ice-covered transmission lines based on conductor end displacement, which can aptly capture the stress characteristics of transmission lines in frozen rain environments. Due to limitations in monitoring data and conditions, these methods face significant constraints. In recent years, visual image-based recognition technology has advanced rapidly. Most research on ice recognition has focused on sea ice, river ice, and road ice (Liu et al., 2025; Ansair et al., 2024; Gui et al., 2023), achieving excellent detection performance. However, studies on ice type recognition for transmission lines remain scarce. This is partly due to the challenges associated with capturing ice images of transmission lines and partly due to the interference caused by complex background noise in such images, which must be accounted for in recognition processes.

Beyond ice physical parameters and imagery, the accuracy of transmission line ice detection can be further enhanced by incorporating environmental data. Numerous studies have demonstrated that meteorological factors, such as wind and humidity, are closely correlated with conductor icing (Dong et al., 2022; Meng et al., 2025; Han et al., 2024), offering valuable insights for ice thickness detection. Therefore, to address the challenges of low accuracy in ice type recognition and thickness detection for transmission lines, this paper proposes DTL-IceNet (Dual-Task Learning Ice Detection Network), a dual-task learning framework designed to enhance the performance of both ice coating recognition and thickness detection. DTL-IceNet employs a multi-branch ice coating recognition module to separately extract spatial feature information of both the background and ice-covered regions, thereby determining the ice type. Simultaneously, a multi-scale attention-based semantic segmentation module is utilized to segment the ice-covered areas. Finally, the model integrates ice type recognition, ice segmentation results, and key meteorological factors to optimize ice thickness estimation, yielding more accurate identification of ice types and thickness on transmission lines. The main contributions of this paper are as follows:

- (1) To address the issue of low ice thickness detection accuracy caused by the irregular shape of ice on transmission lines and complex environmental conditions, this paper proposes a dual-task learning framework, DTL-IceNet. The framework enhances ice thickness detection performance by leveraging ice type classification and key meteorological elements to assist ice segmentation. The proposed framework incorporates an ice coating recognition module, ResSepNet (Residual & Depth-Separable Convolution Network), an icing region segmentation module, MOMSA-SegNet (MobileOne & Multi-Scale Attention Network), and an ice thickness optimized calculation module. By integrating ice segmentation results with ice types and key meteorological factors through multi-scale fusion, the framework refines ice thickness estimation. Through the fusion of multi-source heterogeneous data and the multi-scale fusion of image classification and segmentation techniques, the reliance on a single ice contour for thickness estimation is eliminated, significantly enhancing detection accuracy.
- (2) To address the challenge of incomplete information extraction and utilization in transmission line ice images due to background noise interference, such as fog and light noise, a ResSepNet ice coating recognition module is developed. This module integrates a nested residual structure and depthwise separable convolution to segment the ice image into an upper background area and a lower conductor area. Additionally, three branches are designed to

extract features from the entire image, background, and conductors separately, effectively mitigating the impact of background noise.

(3) Considering the limitations of two-dimensional ice images in representing the spatial distribution of three-dimensional ice, which may lead to detection errors, this study designs the MOMSA-SegNet icing region segmentation module. The module incorporates an improved MobileOne encoder and a multi-scale attention mechanism to segment the ice region from both the main and side perspectives of the image, thereby enhancing the information capture capability of a single perspective. Additionally, a skip connection structure and multi-scale attention mechanism are employed to comprehensively extract spatial features from the raw icing image, further improving segmentation accuracy.

#### 2 Method

The detection of ice in transmission line images primarily involves two tasks: ice type recognition and ice thickness detection. This paper presents DTL-IceNet, a dual-task learning framework for ice detection, designed to achieve ice type recognition and ice thickness detectio for transmission lines. Through the meticulous design of various modules, the proposed framework effectively addresses the limitations in the accuracy of ice type recognition and thickness detection. The overall framework structure of DTL-IceNet is illustrated in Fig. 1.

Figure 1. DTL-IceNet overall framework structure.

Note. For details on the ice coating recognition module, please refer to Section 2.2; for details on the icing region segmentation module, please refer to Section 2.3; for details on the equivalent thickness optimized calculation module, please refer to Section 2.4.

DTL-IceNet primarily consists of three components: the ice coating recognition module (ResSepNet), the icing region segmentation module (MOMSA-SegNet), and the ice thickness optimized calculation module. In the ResSepNet module, the raw icing image undergoes preprocessing to generate the background subgraph and the iced line subgraph. Along with the full ice-covered graph, three branches are employed to extract features from different

spatial regions, which are then fused to determine the ice type. In the MOMSA-SegNet module, the raw icing image

is processed through a multi-scale attention-based semantic segmentation network to segment the ice-covered region

from both the main view and the side view. In the ice thickness optimized calculation module, the ice coating recognition results and icing region segmentation results are integrated, and key meteorological data is introduced for correction and optimization to obtain equivalent ice cover thickness values, thereby realizing the ice type recognition and thickness detection tasks. The subsequent sections will provide a detailed description of the ResSepNet module, MOMSA-SegNet module, and the ice thickness optimized calculation module.

### 2.1 ResSepNet

The ice coating recognition module, ResSepNet, consists of a background branch, an icing branch, and a global branch. It is capable of recognizing four types of icing: ice-free, glaz, rime, and mixed rime. To mitigate background noise interference, the original image is divided into a background subgraph and an iced line subgraph. The background and ice-covered branches extract features from their respective regions, while the global branch utilizes a transfer learning model to capture the overall ice-covered features of the entire image. By employing a multi-branch structure, ice-covered features at different spatial scales are normalized, fused, and recognized to produce the final recognition result. The model structure of ResSepNet is shown in Fig.2, which mainly includes an ice segmentation preprocessing module and a feature extraction and recognition module.

Figure 2. Model structure of ResSepNet.

### 2.1.1 RDS Convolutional Block

To enhance the feature extraction performance of the model in complex icing scenarios, this paper incorporates multiple RDS convolution blocks into ResSepNet, utilizing a nested residual structure and depthwise separable convolution. These blocks serve as the core feature extraction modules in both the background and icing branches. The structure of the RDS convolution block is illustrated in Fig. 3. The convolution block in the background branch is referred to as B-RDS, while the one in the icing branch is denoted as I-RDS. Both branches adopt similar network architectures (as show in Fig. 2). Figure 3 presents the structure of a single RDS convolution block.

Figure 3. A single RDS convolution block.

The nested residual structure in the RDS convolutional block incorporates skip connections, enabling gradients to propagate directly from shallow layers to deeper layers. This effectively mitigates the gradient vanishing problem while preventing network overfitting and degradation. By employing multiple nested residual blocks, the model captures complex features at deeper levels while preserving shallow features, thereby enhancing its capability to extract intricate features in real-world ice-covered scenarios. On the other hand, the RDS convolution block incorporates depthwise separable convolution, a decomposition method that effectively reduces the number of parameters in convolution operations. This significantly enhances the computational efficiency of the network, resulting in a more compact and responsive model. Its flexibility allows deployment in resource-constrained environments, facilitating distributed processing and real-time computation, making it particularly suitable for transmission line ice detection tasks.

### 2.1.2 Multi-Branch Feature Extraction and Fusion Recognition Module

To mitigate the interference of background noise in ice-covered images, ResSepNet employs a three-branch structure comprising a background branch, a global branch, and an icing branch. By extracting local and global features at multiple scales, it effectively reduces the impact of background noise on recognition performance. The raw icing image undergoes preprocessing to generate a background subgraph in the upper region and an iced line subgraph in the lower region. The background subgraph is fed into the background branch to focus on extracting feature information from the background environment. The iced line subgraph is directed to the icing branch to emphasize the extraction of ice feature information in the transmission line area. Meanwhile, the complete image is directly input into the global branch to capture overall ice feature information. The global branch feature extraction network utilizes EfficientNet-B3 (Tan & Le, 2019) with a migration structure. EfficientNet-B3 achieves a balance between model size and feature extraction capability, ensuring effective feature extraction without excessive computational resource consumption. To adapt to the transmission line icing scenario, the ResSepNet global branch enhances EfficientNet-B3 by incorporating an adaptive output layer. This layer primarily consists of a global average pooling (GAP) layer, a squeeze-and-excitation (SE) module (Hu et al., 2018), a 1×1 convolutional layer, a LeakyReLU activation function, and a fully connected (FC) output layer.

After extracting features from the background branch, icing branch, and global branch, ResSepNet normalizes and sums the ice-covered features output by the three branches to mitigate amplitude differences among features from different branches. This process is mathematically represented by (1):

$$f_{mixed} = \frac{f_{bg}}{\|f_{bg}\|} + \frac{f_{ice}}{\|f_{ice}\|} + \frac{f_{main}}{\|f_{main}\|}$$
(1)

where ||f|| denotes the L2 norm of the feature vector, and  $f_{bg}$ ,  $f_{ice}$ , and  $f_{main}$  represent the output features of the background branch, icing branch, and global branch, respectively.  $f_{mixed}$  represents the final multi-branch fusion output feature, which serves as the icing type recognition result of the transmission line, including ice-free, glaze, rime, and mixed rime.

### 2.2 MOMSA-SegNet

The icing region segmentation module, MOMSA-SegNet, employs the improved MobileOne (Vasu et al., 2023) model as its encoder and incorporates a multi-scale skip connection structure in the decoder. This design forms a semantic segmentation network with a large encoder-small decoder architecture, enabling precise segmentation of the ice-covered regions on transmission lines. The module structure is illustrated in Fig. 4.

Figure 4. MOMSA-SegNet module structure.

To address the issue of information loss resulting from a single perspective, which can reduce ice thickness detection accuracy, MOMSA-SegNet segments the transmission lines from both the main and side perspectives in the raw icing image. This segmentation leverages the multi-split transmission line structure to capture ice information more comprehensively. The definitions of the main perspective line and side perspective line in the raw icing image are illustrated in Fig. 5.

Figure 5. Schematic diagram of transmission line from different perspectives.

### 2.2.1 Improved MobileOne Encoder

MobileOne employs a re-parameterized convolutional structure, enabling the transformation of complex branched architectures into a single, efficient convolutional operation during the inference phase. This significantly reduces computational overhead and inference latency. Furthermore, MobileOne is designed with hardware adaptability in mind, ensuring efficient execution on low-power devices. This feature is particularly crucial for edge devices, such as pole tower ice monitoring systems, where model deployment is required. Moreover, the convolutional structure of the MobileOne model exhibits strong capability in capturing local details, making it well-suited for the fine segmentation of ice-covered regions. Given the complexity of ice-covered images of power transmission lines—caused by factors such as lighting variations, haze, and background clutter—this study enhances the original MobileOne by enlarging the dilation rate in its feature encoding module to expand the receptive field (see the left side of Fig. 4). Additionally, multi-scale features are extracted from multiple intermediate layers. By integrating a multi-head attention mechanism, a multi-scale skip connection structure is designed to provide the decoder with contextual spatial features at different scales, thereby enhancing segmentation accuracy in complex ice-covered scenarios.

## 2.2.2 Multi-scale Attention Decoder

The multi-scale attention decoder primarily consists of multiple multi-head self-attention (MHSA) sub-modules, convolutional layers, and upsampling layers. It extracts feature maps from various intermediate layers of the improved MobileOne encoder, as illustrated in Fig. 4. Each feature map is first processed by an MHSA sub-module, after which the self-attention output features are concatenated with the corresponding decoder layer at the same feature scale, thereby forming the multi-scale attention decoder structure. The architecture of the MHSA sub-module is depicted in Fig. 6.

The detailed feature processing procedure of the MHSA sub-module can be expressed by equation (2):

$$output = X + Conv(Concat(soft \max(\frac{Q_i K_i^T}{\sqrt{d_k}})V_i)_{i=1}^k W^o)$$
(2)

where *output* represents the output feature, while X denotes the input feature, which undergoes a linear transformation to obtain the query (Q), key (K), and value (V) matrices. Q, K, and V are divided into h heads, with

each head having its own transformation parameters  $Q_i$ ,  $K_i$ ,  $V_i$ . The attention weight matrix is computed using function soft max, where  $\frac{1}{\sqrt{d_k}}$  serves as a scaling factor to prevent gradient vanishing. The attention weight matrix is then multiplied by  $V_i$  to obtain the output for each head, denoted as soft max( $\frac{Q_i K_i^T}{\sqrt{d_k}}$ ) $V_i$ . Subsequently, the attention outputs of all heads are concatenated using function Concat, followed by a linear transformation  $W^o$  that remaps the transformed features back to the original feature space. To further enhance local feature extraction, MHSA applies an additional convolutional layer (Conv) after the linear transformation, reinforcing the

Figure 6. MHSA submodule.

242243

 $\frac{240}{241}$ 

The multi-scale attention decoder integrates the features from each MHSA output with the original input X through multiple residual structures. This approach preserves the original input information, enhances the model's capability to extract contextual features, and improves its overall stability.

### 2.3 Ice Thickness Optimized Calculation Module

model's capability to extract fine-grained local features.

The dual-task learning framework proposed in this paper simultaneously outputs both ice type and ice thickness. The output of the ice coating recognition module serves not only as a final result but also as a key input for ice thickness estimation. The ice thickness optimized calculation module first performs an initial ice thickness estimation based on the identified ice type and segmentation results. Subsequently, key meteorological data are incorporated to refine the calculation, yielding an optimized ice thickness. Given that actual ice accumulation on transmission lines is typically uneven and irregularly shaped, the equivalent ice cover thickness is adopted as the final representation in the calculation.

In the preliminary estimation of ice thickness, it is essential to determine the major and minor axes of the ice-

covered cross-section. First, the pixel area of the ice-covered region in both the main view and side view of the

263 264

original ice image is obtained based on the segmentation results from MOMSA-SegNet. Given the known wire diameter, the major and minor axes of the ice-covered cross-section can be estimated by comparing the pixel area of the bare wire in the same transmission line under an ice-free condition. The parameters of the ice-covered cross-section are illustrated in Fig. 7, where *d* represents the bare wire diameter, and *a* and *b* denote the major and minor axes of the ice-covered cross-section, respectively.

Figure 7. Schematic diagram of ice cross-section parameters.

Based on the icing type results from the ice coating recognition module, the corresponding icing density can be determined. Subsequently, the icing density is combined with the major and minor axes of the icing cross-section to perform an initial estimation of the equivalent ice cover thickness.

### 2.3.1 Calculation of Equivalent Ice Cover Thickness

According to the layout specifications of overhead transmission lines, the main view line and the side view line are positioned on the same horizontal plane. Therefore, the major and minor diameters can be determined by analyzing the icing conditions of both lines.

The calculation of the major diameter a is given by (3):

$$a = \frac{\sum_{x_{ice}=1}^{W} \sum_{y_{ice}=1}^{H} S(x_{ice}, y_{ice})}{\sum_{x_{wire}=1}^{W} \sum_{y_{wire}=1}^{H} S(x_{wire}, y_{wire})} \times d$$
 (3)

where S(x, y) denotes the pixel value at coordinate (x, y) in the segmentation result generated by MOMSA-

SegNet.  $\sum_{x_{ice}^A=1}^W \sum_{y_{ice}^A}^H S(x_{ice}^A, y_{ice}^A)$  represents the total number of pixels in the segmented ice-covered area, while

 $\sum_{x_{wire}^A=1}^W \sum_{y_{wire}^A=1}^H S(x_{wire}^A, y_{wire}^A)$  denotes the total number of pixels in the bare wire area without ice. The minor axes b can

be computed using the same approach.

$$T = \sqrt{\frac{\rho}{3.6}(ab - d^2) + \frac{d^2}{4} - \frac{d}{2}}$$
 (4)

- Based on the ice type identification result, the corresponding ice density  $\rho$  (Li et al.,2016) is determined.
- According to (4), the irregular ice cross-section can be approximated as a regular circular cross-section with an
- equivalent area, enabling the calculation of the equivalent ice thickness T.

### 2.3.2 Optimization Calculation of Ice Thickness

- Due to factors such as the placement of the ice monitoring device, the shooting angle, and variations in ambient light
- intensity, ice thickness estimates derived solely from ice images often exhibit certain errors. To address this issue,
- this study incorporates meteorological data in addition to ice images, leveraging key surrounding meteorological
- factors to refine and optimize the initial ice thickness calculations. This approach ensures greater alignment with the
- actual freezing conditions and enhances the overall robustness of the algorithm.
- This study maps the latitude, longitude, and image capture time recorded by the ice monitoring device to the
- corresponding environmental meteorological data from ERA5. This mapping enables the extraction of key
- environmental factors, including temperature T (°C), relative humidity H (%), wind speed V (m s<sup>-1</sup>), and
- precipitation P (mm h-1) (Xu et al., 2023), for each ice image. To refine the ice thickness estimation, a key
- meteorological correction factor is constructed using the Gradient Boosting Decision Trees (GBDT) algorithm and
- parameterized as shown in (5):

$$f(T,H,V,P) = \exp[(-\alpha T) \cdot (1+\beta H) \cdot (1-\gamma V) \cdot (1+\delta P)]$$
 (5)

- where  $\alpha$ ,  $\beta$ ,  $\gamma$  and  $\delta$  are correction parameters.  $(-\alpha T)$  indicates that an increase in temperature leads to a
- decrease in ice cover,  $(1 + \beta H)$  indicates that ice coverage increases with increasing humidity,  $(1 \gamma V)$  signifies
- that high wind speed may result in a decrease in ice cover, and  $(1+\delta P)$  indicates that higher precipitation leads
- to greater ice thickness.
- The correction parameters in (5) are determined using the GBDT algorithm. The environmental meteorological
- factors are used as input features, with the optimized ice thickness value serving as the output target variable. The
- input feature set W = [T, H, V, P] and the target variable set  $Y_{true}$  are constructed. After normalizing the input
- feature set W, the GBDT regression model is built, and  $Y_{true}$  is fitted. Assuming that the model's prediction value
- $F_0(w)$  is the mean value of the target variable, the pseudo residual value during the m -th iterative optimization
- process of the regression model can be expressed by (6):

$$r_m^{(i)} = -\frac{\partial L(Y_{true}^{(i)}, F_{m-1}(w))}{\partial F_{m-1}(w)}, i = 1, 2, ..., N$$
 (6)

where N denotes the total number of samples and L represents the MSE loss function.

- The pseudo residual value  $r_m^{(i)}$  is used as the target variable to fit the decision tree and obtain the regression tree
- h(w). The model update is expressed as:

$$F_m(w) = F_{m-1}(w) + \xi \cdot h_m(w)$$
 (7)

- where  $\xi$  represents the learning rate, and h(w) denotes the output of the m-th regression tree. After training,
- the optimized prediction model is obtained, and each correction parameter can be determined using (8):

$$\alpha = \frac{\partial F_M(w)}{\partial T}, \beta = \frac{\partial F_M(w)}{\partial H}, \gamma = \frac{\partial F_M(w)}{\partial V}, \delta = \frac{\partial F_M(w)}{\partial P}$$
 (8)

- where  $\frac{\partial F_M(w)}{\partial w}$  represents the sensitivity of the model prediction to the key meteorological factors. The model
- hyperparameters are adjusted based on accuracy requirements, and the correction process is iteratively optimized to
- obtain the final key meteorological factor correction parameters. The optimized result for the equivalent ice
- thickness  $T_f$  is given by (9):

$$316 T_f = T \cdot f(T, H, V, P) (9)$$

### 317 **3 Experiments**

- This paper constructs a dataset using raw icing images provided by the power grid and conducts performance
- validation experiments on ice coating recognition and ice thickness detection algorithms. The related work primarily
- involves constructing datasets for ice coating recognition and icing region segmentation, training and testing models
- based on these datasets, evaluating model performance, and conducting comparative analyses with existing methods.
- The experiments were conducted on a Windows 11 operating system equipped with an NVIDIA GeForce RTX 3090
- GPU and 24 GB of memory. The proposed model was developed and tested using the PyTorch framework, followed
- by related experiments.

## 3.1 Experimental Plan and Evaluation Indicators

- To evaluate the effectiveness of the proposed method, experiments were conducted on ice coating recognition, ice
- region segmentation, and ice thickness detection. The experimental plan includes: 1) Ablation studies to assess the
- contribution of each branch in the ice coating recognition model, ResSepNet. 2) Performance comparison of
- ResSepNet with other mainstream classification models for ice type recognition. 3) Segmentation performance
- comparison between MOMSA-SegNet and other advanced segmentation models. 4) Transmission line icing state
- detection in real-world scenarios using the proposed DTL-IceNet model.
- To facilitate model training and testing for ice coating recognition and ice thickness detection, this paper utilizes
- ice monitoring images captured by ice-viewing devices deployed in the power grid. Corresponding datasets are
- constructed based on ice coating recognition and ice thickness detection tasks to meet the training, validation, and
- testing requirements of the proposed algorithm.

- For the ice type recognition experiment, this paper primarily evaluates and compares model performance using
- classification accuracy, precision, recall, F1-score, and the confusion matrix. The calculation formulas for each
- metric are as follows:

$$Accuracy = \frac{TP + TN}{TP + TN + FP + FN}$$
 (10)

$$340 precision = \frac{TP}{TP + FP} (11)$$

$$341 recall = \frac{TP}{TP + FN} (12)$$

$$F1-Score = \frac{2 \cdot precision \cdot recall}{precision + recall}$$
(13)

- where TP denotes the number of samples correctly classified as positive, TN denotes the number of samples
- correctly classified as negative, FP denotes the number of samples incorrectly classified as positive, and FN
- denotes the number of samples incorrectly classified as negative.
- For the icing region segmentation experiment, this paper primarily employs intersection over union (IoU), mean
- IoU (MIoU), and mean pixel accuracy (mPA) to assess and compare the segmentation performance of the model.
- The formulas for each metric are as follows:

$$IoU = \frac{|A \cap B|}{|A \cup B|} \tag{14}$$

$$MIoU = \frac{1}{N} \sum_{i=1}^{N} IoU_i$$
 (15)

$$mPA = \frac{1}{N} \sum_{i=1}^{N} \frac{P_i}{T_i}$$
 (16)

- where A represents the predicted target area, B represents the actual target area,  $A \cap B$  denotes the
- overlapping area between the two,  $A \cup B$  denotes their total coverage area,  $P_i$  represents the number of correctly
- classified pixels for category i, and  $T_i$  represents the total number of actual pixels in category i.

## 355 **3.2 Dataset**

- For the tasks of ice coating recognition and icing region segmentation, this paper constructs two datasets. The ice
- coating recognition task focuses on classifying different types of icing on transmission lines. Therefore, a diverse set
- of icing samples, including ice-free, glaz, rime, and mixed rime, is selected from a large collection of original
- transmission line icing images. During the data preprocessing stage, manual labeling is employed to classify each
- sample, ensuring label accuracy and consistency. Subsequently, data cleaning is performed to remove blurry,
- abnormally captured, or incomplete images, retaining only clear and representative ice-covered samples. Finally, an

ice coating recognition dataset, IceType, consisting of 20,684 images, was constructed and divided into a training set, test set, and validation set in a 6:2:2 ratio.

The ice thickness detection task is primarily accomplished through semantic segmentation, focusing on pixel-level recognition of ice-covered and background areas in transmission line images. Based on the raw icing images, this study manually selects high-quality images with clearly distinguishable ice-covered regions. Subsequently, the ImageLabeler tool in Matlab is used to label the ice-covered areas pixel by pixel, ensuring that each pixel's category label accurately corresponds to either the ice-covered region or the background. Meanwhile, considering the fixed shooting angle characteristics of the ice-viewing device, the ice-covered images underwent appropriate preprocessing and cropping. To enhance data volume and enrich sample distribution, random flipping was applied. Ultimately, an ice-covered region segmentation dataset, IceSeg, containing 6,360 pixel-level annotations, was constructed and split into training, test, and validation sets in a 7:2:1 ratio.

### 3.3 Ice Type Identification Experiment

#### 3.3.1 ResSepNet Branch Ablation Experiment

To evaluate the performance of each branch in ResSepNet, this study conducts a controlled experiment comparing ResSepNet with its individual branches on the IceType dataset. The accuracy and loss variation curves of ResSepNet and its branches on the IceType validation set are illustrated in Fig. 8.

Figure 8. (a) validation accuracy and (b) validation loss. Performance of each branch of ResSepNet.

As shown in Fig. 8, the background branch alone yields suboptimal ice type recognition performance, achieving an accuracy of only 86.55 %. The ice branch improves recognition accuracy to 89.82 %; however, it remains insufficient due to the omission of environmental factors. ResSepNet, which integrates the background branch, icing branch, and global branch, comprehensively accounts for both environmental influences and transmission line icing characteristics, ultimately achieving a recognition accuracy of 95.23 %.

To more clearly illustrate the performance contribution of each branch across different ice types, the confusion matrix for ice type recognition on the IceType test set is presented in Fig. 9. Based on the confusion matrix, it can be observed that the recognition accuracy of each branch and ResSepNet for mixed rime is lower compared to other ice types. This is attributed to the complex morphology of mixed rime. Nevertheless, ResSepNet still achieves a high recognition accuracy of 89.74 % for this type. This is because ResSepNet simultaneously extracts multi-scale

 $\frac{400}{401}$ 

features from the background area, ice-covered area, and the entire image, enabling a more comprehensive capture of image information. The background branch demonstrates superior rime recognition compared to other branches due to the distinct color differentiation of this type. The icing branch excels in recognizing bare wire (ice-free) since the morphology of the wire in this category exhibits more significant differences. The global branch maintains a more balanced recognition across various types, as it does not specifically extract local area features. This also compensates for the limitations of the background branch and icing branch in recognizing mixed rime and other complex types. Overall, ResSepNet achieves outstanding performance in ice type recognition, attaining high accuracy, which confirms that the multi-branch design is well-structured and significantly enhances the recognition capability for ice-covered types.

**Figure 9.** Confusion matrix of ice type recognition effect of each branch of ResSepNet.

## 3.3.2 Comparative Experiments with ResSepNet

To evaluate the ice coating recognition performance of ResSepNet, comparative experiments were conducted on the IceType dataset using mainstream models such as EfficientNet-V2 (Tan & Le, 2021), MobileNet-V3 (Howard et al., 2019), ResNeXt (Xie et al., 2017), and MobileOne (Vasu et al., 2023). The accuracy and loss curves for each model on the IceType validation set are presented in Fig. 10.

 $\frac{407}{408}$ 

Figure 10. (a) validation accuracy and (b) validation loss. Performance comparison of various models.

As shown in Fig. 10, the ResSepNet model proposed in this paper not only achieves the highest accuracy but also demonstrates superior convergence speed and stability. This performance can be attributed to the model's lightweight and multi-branch structure, which allows it to maintain a compact size while ensuring rapid convergence. Additionally, the multi-branch design enables more comprehensive capture of the ice coverage information, reduces the impact of background noise, and enhances the overall recognition accuracy. The following section presents a comparison of various evaluation metrics for each model on the IceType test set, as shown in Fig. 11.

Figure 11. Comparison of evaluation indicators of various models.

From the radar chart comparison results in Fig. 11, it is evident that the proposed ResSepNet outperforms other models in terms of accuracy, precision, recall, and F1-score. Compared with other methods, the proposed method achieves an average improvement of 4.17 % in accuracy, 4.79 % in weighted precision (W-Prec), 4.17 % in weighted recall (W-Recall), and 4.28 % in weighted F1-score (W-F1). Additionally, macro precision (M-Prec), macro recall (M-Recall), and macro F1-score (M-F1) exhibit average improvements of 4.55 %, 4 %, and 4.26 %, respectively. Combined with the results in Fig. 10, these findings demonstrate that ResSepNet consistently maintains superior performance in ice type recognition, both in terms of individual evaluation metrics and overall effectiveness. The specific values corresponding to Figs. 10 and 11 are presented in Table 1, where the bolded values indicate the best results.

Table 1.Performance comparison results of each model

| Modules         | Accuracy(%) | W-Prec(%) | W-Recall(%) | W-F1  | M-Prec(%) | M-Recall(%) | M-F1  |
|-----------------|-------------|-----------|-------------|-------|-----------|-------------|-------|
| EfficientNet-V2 | 86.87       | 87.65     | 86.87       | 86.57 | 84.71     | 84.41       | 82.57 |
| MobileNet-V3    | 90.93       | 90.86     | 90.93       | 89.67 | 87.13     | 85.55       | 84.42 |
| ResNeXt         | 93.02       | 93.83     | 93.02       | 92.80 | 90.10     | 90.69       | 90.05 |
| MobileOne       | 93.41       | 94.26     | 93.41       | 93.21 | 91.21     | 90.71       | 89.27 |
| ResSepNet(Ours) | 95.23       | 96.44     | 95.23       | 94.84 | 92.84     | 91.84       | 90.84 |

### 3.4 Icing Region Segmentation Experiment

## 3.4.1 Segmentation Effects in Different Scenarios

Since the accuracy of ice thickness calculation is directly influenced by the segmentation results of the icing region, this study evaluates the performance of the proposed icing region area segmentation module under various environmental conditions. To this end, segmentation tests were conducted in representative scenarios, including sunny days, heavy fog, and nighttime. The results are presented in Fig. 12, where the red regions indicate the segmentation results for the main view line, while the yellow regions represent those for the side view line.

Figure 12. Icing region segmentation results of MOMSA-SegNet in different scenarios.

As illustrated in Fig. 12, the proposed icing region segmentation module, MOMSA-SegNet, effectively segments both the main view line and the side view line across different environmental conditions, including sunny days, foggy conditions, and nighttime. These results demonstrate that the proposed segmentation method can reliably meet the requirements for ice thickness calculation.

### 3.4.2 Comparison of Segmentation Performance of Different Models

To further evaluate the segmentation performance of the proposed MOMSA-SegNet, classic models such as UNet++ (Zhou et al., 2018), SegNet (Badrinarayanan et al., 2017), and DySample (Lin et al., 2017) were trained on the IceSeg dataset and compared with MOMSA-SegNet on the test set. The evaluation primarily focused on key metrics, including the Intersection over Union (IoU) for the main view, side view, and background, as well as the mean IoU (MIoU) and mean Pixel Accuracy (mPA). The comparative results are presented in Table 2.

Table 2. Comparison of segmentation performance of different models

| Table 2. Comparison of segn | remation performant | ee of afficient models |                |       |       |
|-----------------------------|---------------------|------------------------|----------------|-------|-------|
| Modules                     | main view IoU       | Side view IoU          | background IoU | MIoU  | mPA   |
|                             | (%)                 | (%)                    | (%)            |       |       |
| UNet++                      | 86.12               | 70.97                  | 98.73          | 85.27 | 64.85 |
| SegNet                      | 84.43               | 75.22                  | 98.79          | 86.15 | 65.80 |
| DySample                    | 85.25               | 76.63                  | 98.89          | 87.83 | 66.63 |
| MOMSA-SegNet(Ours)          | 86.17               | 79.05                  | 98.96          | 88.06 | 67.17 |

From Table 2, it can be observed that although the performance differences among the models are relatively small, MOMSA-SegNet achieves the highest scores across all evaluation metrics. Specifically, compared to other models, the proposed method improves the IoU of the main view and side view by 0.9 % and 4.78 %, respectively. Additionally, it enhances background IoU by 0.16 %, while MIoU and mPA increase by 1.64 % and 1.41 %, respectively. These results highlight the superior segmentation performance of MOMSA-SegNet across different scenarios. The comparative segmentation results of each model on the IceSeg test set are illustrated in Fig. 13.

Figure 13. (a) first picture during the day, (b) second picture during the day, (c) first picture during the night and (d) second picture during the night. Comparison of segmentation effects of different models.

Figure 13 intuitively demonstrates that the segmentation performance of each model on the main view line exhibits minimal differences. However, under the influence of factors such as fog and ambient light, the segmentation results for the side view line vary significantly among models. Notably, the proposed MOMSA-SegNet achieves superior segmentation performance on the side view line and demonstrates the best overall performance. This can be attributed to its jump connection structure and multi-scale attention mechanism, which effectively capture the characteristics of different view lines and provide precise support for subsequent ice thickness calculations.

# 3.5 Ice Thickness Detection Experiment

To verify the accuracy of the final ice thickness measurement, a simple pole tower and conductor device were constructed at the experimental site of Nanjing University of Information Science and Technology. This setup simulated the actual ice conditions of the transmission line in a natural environment. Using an ice viewing device, a small transmission line ice thickness dataset was created, covering ice thickness levels ranging from 0 to 30 mm, with a bare wire diameter of 33.8 mm. To approximate the shooting angle of real ice monitoring equipment, pixel expansion processing was applied to the original images, followed by annotation of the ice-covered areas. The results are shown in Fig. 14. Due to site conditions, no side view line was included. The performance of the proposed model was validated using the thickness dataset.

Figure 14. The segmentation effect of the ice-covered image of the simulated transmission line at the experimental site.

Based on the ice data collected from the observation field, we tracked the complete ice accumulation and variation process of the transmission line over a 24-hour period. The ice thickness measurement results for this process are illustrated in Fig. 15, with detailed values presented in Table 3. In the figure, dark blue represents the actual ice

 thickness, light blue denotes the optimized ice thickness estimated by the proposed model, and green indicates the initial ice thickness of the main (side) view line measured by the model. Given that the simulated ice cross-section is circular, the side view thickness is assumed to be equal to the main view thickness.

As shown in Fig. 15, both the initial and optimized thickness values align with the overall trend of the actual thickness, demonstrating that the proposed model can accurately capture the growth pattern of ice thickness. Moreover, the optimized thickness measurement is closer to the actual thickness, indicating that key meteorological data effectively refine the ice thickness estimation, yielding more accurate measurement results. To further evaluate the accuracy of the proposed method in detecting ice thickness, we compared the optimized and initial ice thickness measurements with the actual values. Additionally, we calculated key evaluation metrics, including the mean absolute percentage error (MAPE), Pearson correlation coefficient (PCC), and mean square error (MSE). The results are presented in Table 4.

Figure 15. Comparison curve between measured value and actual value.

Table 3 Comparison between measured values and actual values

| Table 3. Comparison between measured values and actual values. |            |            |          |          |          |            |            |            |            |            |            |            |
|----------------------------------------------------------------|------------|------------|----------|----------|----------|------------|------------|------------|------------|------------|------------|------------|
| Time                                                           | 1:00       | 2:00       | 3:00     | 4:00     | 5:00     | 6:00       | 7:00       | 8:00       | 9:00       | 10:00      | 11:00      | 12:00      |
| Actual thickness (mm)                                          | 5          | 5          | 5        | 5        | 10       | 10         | 15         | 15         | 20         | 20         | 15         | 15         |
| Initial thickness (mm)                                         | 7.6        | 8.1        | 8.1      | 8.1      | 16.6     | 16.6       | 24.6       | 24.6       | 26.6       | 26.6       | 24.6       | 24.6       |
| Optimized                                                      |            |            |          |          |          |            |            |            |            |            |            |            |
| thickness                                                      | 4          | 4          | 4        | 4        | 11       | 11         | 17         | 17         | 19         | 19         | 17         | 17         |
| (mm)                                                           |            |            |          |          |          |            |            |            |            |            |            |            |
| Time                                                           | 13:00      | 14:00      | 15:00    | 16:00    | 17:00    | 18:00      | 19:00      | 20:00      | 21:00      | 22:00      | 23:00      | 24:00      |
| Actual thickness                                               |            |            |          |          |          |            |            |            |            |            |            |            |
| (mm)                                                           | 10         | 10         | 5        | 5        | 5        | 10         | 15         | 20         | 25         | 30         | 30         | 30         |
| (mm) Initial thickness (mm)                                    | 10<br>17.1 | 10<br>17.1 | 5<br>7.6 | 5<br>8.1 | 5<br>8.1 | 10<br>17.1 | 15<br>24.6 | 20<br>26.6 | 25<br>32.6 | 30<br>44.6 | 30<br>44.6 | 30<br>44.1 |

Note. The two lines of time represent the 12 hours before and after a day.

Table 4. Evaluation index of ice thickness measurement value.

| Evaluation index | MAPE(%) | PCC | MSE |
|------------------|---------|-----|-----|

https://doi.org/10.5194/egusphere-2025-3097 Preprint. Discussion started: 13 November 2025 © Author(s) 2025. CC BY 4.0 License.

499

500501

| Initial ice thickness   | 56.36 | 0.97 | 65.60 |
|-------------------------|-------|------|-------|
| Optimized ice thickness | 11.82 | 0.99 | 1.83  |

As observed in Fig. 15 and Table 4, the optimized ice thickness calculated by the proposed model closely follows the actual values. The Pearson correlation coefficient reaches 0.99, indicating a strong correlation, while the measurement error remains minimal. The mean absolute percentage error is only 11.82 %, and the mean square error is as low as 1.83, demonstrating the model's high accuracy. These results confirm that the proposed method performs well in real-world scenarios and meets the practical application requirements.

### 4 Summarize

To address the challenge of insufficient accuracy in ice coating recognition and thickness detection for high-altitude transmission lines, this paper proposes DTL-IceNet, dual-task learning architecture with multi-scale fusion mechanisms for enhanced ice detection on transmission lines, which enables precise ice coating recognition and ice thickness estimation. The proposed method employs ResSepNet, a multi-branch network designed to fuse and extract ice features across different spatial scales, effectively mitigating background noise interference and enhancing ice type classification accuracy. Additionally, a semantic segmentation network, MOMSA-SegNet, incorporating a skip structure and multi-scale attention mechanism, is utilized to segment icing regions on transmission lines, thereby facilitating ice thickness estimation. Furthermore, key meteorological data are integrated to optimize the correction of ice thickness measurements. Based on the original ice images provided by the power grid, we constructed a series of ice image datasets, including IceType and IceSeg. The experimental results demonstrate that the proposed DTL-IceNet achieves 4.17 % higher ice recognition accuracy compared to EfficientNet-V2, MobileNet-V3, ResNeXt, and MobileOne, while its ice area segmentation MIoU surpasses that of mainstream segmentation models such as UNet++ by 1.64 %. These findings indicate that the dual-task learning framework effectively detects and identifies both ice type and thickness on transmission lines. Furthermore, in the simulation test at the test site, the MAPE of ice thickness estimation reached 11.82 %, and the PCC attained 0.99, demonstrating the proposed method's robust ice detection performance in real-world conditions. However, due to hardware limitations, this study does not account for the impact of terrain elements on transmission line icing. The detection performance of the proposed method under significant environmental changes requires further improvement. Future work will focus on incorporating terrain elements into the model and examining their correlation with transmission line icing.

## **Data Availability Statement**

The datasets and code utilized for the analyses in this study are publicly available at 526 https://doi.org/10.5281/zenodo.15718305 (Fu et al., 2025).

#### 527 Author contributions

- Yufei Fu, Wenjie Zhang planned the campaign; Yang Cheng, SongYuan Cao, Ling Tan performed the
- measurements; Yufei Fu, Jiaxin He, Mengya Wang, Wenjie Zhang analyzed the data; Yufei Fu and Wenjie Zhang
- wrote the manuscript draft; Wenjie Zhang, Yang Cheng, SongYuan Cao, reviewed and edited the manuscript.

### 531 Competing interests

The authors declare that they have no conflict of interest.

### 533 Acknowledgments

- The authors declare no conflicts of interest. This work was supported by National Key R&D Program of China
- (Grant No. 2023YFE0208100).

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
