# Peer review of "DTL-IceNet: A Dual-Task Learning Architecture with Multi- # 2 Scale Fusion Mechanisms for Enhanced Ice Detection on # 3 Transmission Lines"

_EGUsphere, 2025_

## Author Comment (AC1)

Thank you very much for taking the time to review this manuscript. I have responded to each of the issues raised by the two reviewers and have revised the manuscript.

**Response to Reviewer 1 Comments**

Comments 1- It is recommended to supplement the Summarize section or add a dedicated Discussion section. This section should include a more in-depth analysis of the reasons why the model performs well or fails under certain conditions, a discussion of the model's limitations (for example, its performance under significant terrain variations or extreme weather conditions not represented in the dataset), and a more balanced interpretation of the results in the context of existing literature.

**Response:** We sincerely thank the reviewer for this important and constructive suggestion. In response to your comment, we have added a dedicated Section 4 Discussion to provide a more in-depth analysis of the reasons behind the model's performance under different conditions, its potential failure cases, and its proper positioning within the context of existing studies. In the revised Discussion, we analyze at the mechanism level why the proposed model achieves strong performance in ice type recognition, ice region segmentation, and ice thickness estimation, with particular emphasis on the roles of multi-branch feature decomposition, multi-scale attention mechanisms, and the collaborative modeling of geometric and meteorological constraints in improving both discriminative ability and physical consistency. Meanwhile, we explicitly supplement the main limitations of the current work, noting that the existing dataset does not yet cover extreme weather conditions such as strong convection and freezing rain combined with high winds, nor does it systematically represent complex terrains with significant topographical variations; therefore, the model's generalization capability under extreme meteorological and complex terrain conditions still requires further validation. In addition, the revised manuscript strengthens the comparison and connection with existing literature, providing a more balanced and objective interpretation of the results from the perspectives of background sensitivity, weak-boundary segmentation robustness, and meteorology-driven icing processes.

Specific modifications are as follows: **(Page 23-25)**

**4 Discussion**

**4 Discussion**

The dual-task learning framework, DTL-IceNet, proposed in this study demonstrates high reliability in both ice type
recognition and thickness detection tasks. Its main contribution lies in the unified modeling of three types of infor-
mation: type, geometry, and meteorology, which effectively enhances the comprehensive sensing ability of icing con-
ditions on transmission lines. This fusion-based design aligns with the view emphasized in the literature that "ice
physics, image features, and environmental processes must be considered in a coordinated manner" (Fan et al., 2018;
Hao et al., 2023; Chen et al., 2024; Dong et al., 2022; Meng et al., 2025; Han et al., 2024), and it achieves both
discriminative ability and physical consistency in typical monitoring scenarios, leading to significant improvements
over existing methods.

In the ice type recognition task, the model explicitly separates the background, conductor, and icing regions through
a multi-branch feature extraction structure. Studies by Fan et al. (2018). and Hao et al. (2023) have pointed out that
ice recognition is highly sensitive to the environmental background, and deep networks based on a single-path feature
extraction often struggle to fully capture the local texture of the conductor in complex backgrounds. The decomposi-
tion-based modeling approach of DTL-IceNet significantly enhances the distinction between different types of icing
under complex lighting, fog, and noise conditions, providing stronger anti-interference ability compared to single-
branch methods.

In the ice region segmentation task, MOMSA-SegNet leverages a multi-scale attention module to improve the rep-
resentation capability of thin conductors and irregularly shaped icing areas. Existing research has shown that tradi-
tional edge detection or low-level feature methods exhibit poor robustness in weak boundaries, low contrast, and
nighttime scenarios (Han et al., 2024; Tan & Le, 2019; Hu et al., 2018; Vasu et al., 2023; Li et al., 2016). In contrast,
the multi-scale attention mechanism effectively utilizes the contextual structure surrounding the transmission line,
allowing the model to maintain stable geometric contour predictions in typical scenes such as sunny, foggy, and
nighttime conditions. Its accuracy advantage stems from the targeted utilization of the transmission line image struc-
ture rather than relying solely on the depth of the network or the scale of parameters.

For ice thickness estimation, the improvement in model performance is mainly attributed to the synergistic effect
of geometric and meteorological constraints. The image geometric information provides the basic trend of ice volume
changes, but relying solely on 2D images cannot accurately reflect the true 3D shape of the ice, leading to systematic
biases during temperature, humidity, wind speed, and precipitation phases. By introducing the meteorological correc-
tion term based on environmental factors, the model performs consistent corrections on the initial thickness estimation
according to the basic physical laws of ice growth and melting, effectively compensating for the inherent structural
biases in the geometric estimation. Experimental results show that the thickness curves align more closely with the
actual distribution across multiple phases, indicating the complementary role of geometric and meteorological infor-
mation in thickness estimation.

Although DTL-IceNet demonstrates robustness under typical monitoring conditions, its applicability is still limited
by the data coverage and experimental conditions. The data used in this study were primarily collected in typical
meteorological scenarios and have not yet covered extreme weather conditions such as severe convection or freezing
rain coupled with strong winds. Under such conditions, the signal-to-noise ratio of images and the rate of ice mor-
phology change may exceed the training distribution, and the model's robustness needs further validation. Moreover,
the current experimental data do not systematically reflect complex spatial environments with significant topographic
variations. Existing studies have shown that terrain, especially in valley wind fields, significantly impacts icing dis-
tribution, so the model's performance in complex terrain scenarios remains uncertain.

Comments 2- The manuscript presents two distinct results: (1) MOMSA-SegNet achieves the highest segmentation mIoU, and (2) the overall framework reports a thickness MAPE of 11.82%. To clarify the relationship between these two findings, additional controlled experiments would be helpful. In particular, demonstrating that, under the same test set and using the same thickness estimation procedure, the proposed segmentation model yields a consistently lower thickness estimation error compared with the other segmentation models discussed in the paper would provide more direct evidence of its contribution. Without such comparisons, the extent to which the segmentation module influences the final estimation accuracy remains uncertain.

**Response:** We sincerely thank the reviewer for this targeted and insightful comment. In response to your concern that the independent presentation of segmentation accuracy and final thickness accuracy is insufficient to directly demonstrate the actual contribution of the front-end segmentation module to thickness estimation, we have added a new Section 3.5.2, entitled "Effect of Different Segmentation Models on Ice Thickness Detection," and included Table 5 in the revised manuscript. This new experiment systematically compares the impacts of UNet++, SegNet, DySample, and the proposed MOMSA-SegNet on thickness estimation accuracy under the same observation-site dataset and identical ice-type recognition and meteorological correction processes. The results indicate that, under fully consistent thickness calculation and meteorological correction procedures, better segmentation performance leads to lower MAPE and MSE for both the initial and optimized thickness estimates, and MOMSA-SegNet achieves the lowest errors in both cases. Moreover, even after introducing meteorological correction, the error differences among different segmentation models remain evident, indicating that geometric structural biases caused by segmentation errors cannot be completely eliminated by meteorological correction alone. These findings directly verify, under a unified experimental framework and dataset, the substantial influence of the segmentation module on the final thickness estimation accuracy and clearly demonstrate the fundamental and irreplaceable role of segmentation quality in thickness estimation. We sincerely appreciate your professional guidance and valuable suggestions, which have significantly strengthened the rigor and completeness of this work.

Specific modifications are as follows: **(Page 21-22)**

**3.5.2 Effect of Different Segmentation Models on Ice Thickness Detection**

**3.5.2 Effect of Different Segmentation Models on Ice Thickness Detection**

To quantitatively evaluate the impact of the segmentation module on the accuracy of final ice thickness estimation, this study, under the premise of maintaining consistency in the ice type recognition and thickness optimization calcu- lation processes, uses UNet++, SegNet, DySample, and MOMSA-SegNet as the frontend segmentation models. The thickness estimation experiments were conducted on the same observation field test dataset. The ice type recognition module and thickness optimization calculation module were kept unchanged, and only the ice segmentation submod- ule was replaced with different typical methods. The initial thickness and optimized thickness errors were then com- puted on the corresponding observation field thickness test data. This approach allows a direct comparison of the effects of different segmentation models on geometric scale calculation and error propagation within a unified frame- work, providing a clearer insight into the structural relationship between segmentation quality and thickness detection accuracy. The experimental results are shown in Table 5.

**Table 5.** Comparison of ice thickness detection performance driven by different segmentation models

| Modules | Initial ice thickness MAPE(%) | Optimized ice thickness MAPE(%) | Initial ice thickness MSE | Optimized ice thickness MSE |
|---|---|---|---|---|
| UNet++ | 69.53 | 14.58 | 80.93 | 2.26 |
| SegNet | 65.38 | 13.71 | 76.09 | 2.12 |
| DySample | 57.45 | 12.05 | 66.86 | 1.87 |
| **MOMSA-SegNet(Ours)** | **56.36** | **11.82** | **65.60** | **1.83** |

As seen in Table 5, under the same ice type recognition and meteorological correction processes, the performance of ice thickness detection shows a consistent trend with different segmentation models as frontend submodules. The better the segmentation performance, the lower the initial and optimized thickness errors. When MOMSA-SegNet, the segmentation model proposed in this paper, is used, both the MAPE and MSE of the initial thickness are the lowest.

After replacing it with UNet++, SegNet, or DySample, both errors increase to varying degrees. This indicates that the segmentation stage directly affects the accuracy of the ice contour and area depiction, which in turn influences the downstream geometric parameter estimation. Segmentation errors accumulate and amplify in the calculation of equiv- alent ice thickness.

A further comparison of the initial and optimized thickness metrics reveals that after the introduction of meteoro-
logical correction, the MAPE and MSE for all models significantly decrease, showing that the environmental-driven
correction terms can effectively compensate for the system errors caused by the 2D perspective and geometric simpli-
fications. However, the relative differences between the models still remain even after optimization. Even with mete-
orological correction, the thickness estimation based on MOMSA-SegNet maintains the lowest MAPE and MSE,
while segmentation models with weaker performance still exhibit higher optimization errors. This suggests that the
meteorological correction mainly targets global system biases related to environmental processes, and cannot fully
counteract the structural errors in the ice region contour and scale caused by segmentation. Thus, it can be concluded
that segmentation quality determines the geometric baseline for thickness estimation, while meteorological correction
fine-tunes this baseline, forming a hierarchical complementary relationship.
In conclusion, under the condition of complete consistency in the thickness estimation algorithm and test dataset,
simply replacing the segmentation module results in a monotonous or nearly monotonous decrease in thickness errors
with the improvement of segmentation model performance. This result confirms the substantial contribution of
MOMSA-SegNet in ice thickness detection from a data-driven perspective. Its higher segmentation accuracy not only
reflects in pixel-level metrics but also significantly reduces the initial errors in downstream thickness estimation,
maintaining its advantage even after meteorological correction and effectively transmitting the improved segmentation
performance to the final physical quantity estimation results.

Comments 3- Regarding environmental conditions, although the segmentation component includes descriptions of performance under different weather scenarios, the influence of these varying conditions on the final thickness estimation results is not examined. Expanding the Discussion section to include an evaluation of thickness estimation performance across different weather conditions would help provide a more complete understanding of the method's behavior.

**Response:** We sincerely thank the reviewer for the constructive suggestions regarding the related work. In response to your comment, we have added a specific analysis in Section 4 Discussion on the potential influence of different meteorological conditions on thickness estimation. We clearly state that, due to the practical limitations of the observation site, the current thickness validation experiments only include real thickness calibration data under daytime clear-sky conditions, and thus a direct quantitative comparison of thickness errors under multiple weather conditions is not yet available. On this basis, combined with the segmentation results and the physical mechanism of icing formation, we systematically analyze at the mechanism level two main pathways through which weather factors may affect thickness estimation: on the one hand, imaging degradations such as heavy fog and low illumination weaken the clarity of the ice–conductor boundaries, thereby influencing the geometrically derived initial thickness through segmentation errors; on the other hand, meteorological variables such as temperature, humidity, and wind speed strongly drive the growth and ablation of icing and introduce systematic corrections to thickness estimation through the meteorological correction module. We also objectively point out that under severe imaging degradation, meteorological correction cannot completely offset the amplification of geometric errors, and therefore the thickness estimation accuracy under real complex weather conditions still requires further investigation. Once again, we sincerely appreciate the reviewer's valuable comments, which have played an important role in improving the completeness and rigor of this work.

Specific modifications are as follows: **(Page 24-25)**

**4 Discussion**

While the segmentation experiments presented in this study systematically show performance differences under
various imaging conditions such as sunny days, fog, and nighttime, the thickness validation experiments were limited
by the actual conditions of the observation field, which only included real thickness calibration data from sunny sce-
narios and could not directly quantify thickness estimation accuracy under various weather conditions. To address this
logical gap, we will further analyze the potential impact of weather factors on thickness estimation. Mechanistically,
weather changes affect thickness results mainly through two paths. First, imaging degradation such as fog and low
light reduces the clarity of the ice and conductor boundaries, causing segmentation masks to deviate in geometric
details and directly affecting the preliminary thickness calculation based on area and contour inference. Second, me-
teorological variables such as temperature, humidity, and wind speed determine the growth and melting rates of the
ice, strongly driving the temporal evolution of thickness. The meteorological correction module of DTL-IceNet can
provide systematic corrections based on this driving pattern, but it cannot completely compensate for geometric devi-
ations caused by severe imaging degradation. Therefore, under real fog or nighttime conditions, the initial thickness
errors may increase, while the optimized thickness is expected to show more stable but still limited corrections.

Comments 4- The manuscript claims to propose a comprehensive "dual-view" solution. While the approach of collecting real thickness data in controlled field experiments is understandable and commendable, the current experimental setup does not sufficiently validate the core contribution of the method, and instead highlights certain limitations. Specifically, the final performance evaluation is conducted on a restricted, single-view version of the system. This creates a substantial mismatch between the claimed capabilities and the empirical validation. We note that the authors provide "site conditions" as a rationale for this choice. However, this results in the core claim of the method—that leveraging multi-view structures from a single image enhances information capture—remaining unverified in thickness estimation experiments. By effectively omitting the higher-error components during validation, a critical question arises: does the reported thickness accuracy truly reflect the capability of the complete main-view and side-view system, or does it primarily represent performance in a simplified main-view scenario, which conveniently avoids the error propagation associated with the less accurate side-view segmentation?

**Response:** We sincerely thank the reviewer for the valuable comments and suggestions regarding the proposed "dual-view" framework. We fully understand and agree with the core concern you raised. Due to the fact that the current experimental conditions only provide real data from the main-view camera, and no true side-view data are available at present, the icing cross-section is assumed to be circular in our simulations; therefore, the side-view thickness is set equal to the main-view thickness for the purpose of calculation. In response to your comment, we have added a dedicated discussion in Section 4 to explicitly clarify and reflect on the experimental validation boundary of the dual-view scheme. We clearly state that, constrained by the observation-site layout and transmission line installation conditions, the current thickness validation experiments do not yet include transmission lines with real side-view imaging, and the thickness estimation therefore relies on a simplified geometry dominated by the main-view assumption. As a consequence, the reported thickness accuracy primarily reflects the performance under a single-view scenario rather than the theoretical upper bound of the complete dual-view structure. We also objectively point out that the error propagation effect introduced by side-view segmentation has not yet been quantitatively evaluated, which constitutes an important limitation of the current engineering validation. Nevertheless, the existing results still verify the feasibility and effectiveness of the proposed geometry–meteorology fusion framework through the main-view pathway and provide a solid foundation for further validation of the dual-view structure in more complex scenarios. We have accordingly identified the complete experimental validation of the dual-view scheme as a major direction for future work and plan to construct a real multi-view acquisition platform to systematically evaluate the performance gain of multi-view geometric constraints and the associated error propagation mechanisms in thickness estimation.

Specific modifications are as follows: **(Page 25)**

**4 Discussion**

The dual-perspective approach proposed in this study is one of the key innovations of the overall framework.
Through joint segmentation of the main and side perspectives, the major and minor axes of the ice-covered cross-
section can be theoretically estimated, improving the certainty of geometric parameters. However, it is important to
note that the thickness validation experiment in this paper was limited by site constraints and did not actually deploy
transmission lines for real side-perspective imaging. Thickness detection relied on a simplified geometric assumption
primarily based on the main perspective. This experimental condition means that the final obtained thickness accuracy
mainly reflects the performance under the simplified single-perspective system rather than the upper limit of a com-
plete dual-perspective structure. Therefore, while the current results prove the feasibility and potential value of the
proposed framework, they do not fully validate the information gain of the dual-perspective structure in real multi-
conductor scenarios. This also suggests that the side-perspective segmentation error's impact on thickness calculation
has not been fully quantified. Future work will focus on building an experimental platform that truly reflects the dual-
perspective structure to systematically evaluate error propagation mechanisms and further optimize geometric fusion
methods.

**Comments 5-** On the other hand, achieving strong final results does not, by itself, validate the correctness or effectiveness of the front-end image segmentation plus area ratio approach. It primarily demonstrates the strength of the back-end correction module. Only with supplementary ablation experiments can it be convincingly shown that meteorological data and image information are complementary and both necessary, thereby substantiating the true value of the fusion framework.

**Response:** We sincerely appreciate the reviewer's careful and insightful comments. We fully agree with your concern that strong final thickness estimation results alone are insufficient to validate the effectiveness of the front-end image segmentation and area-ratio-based geometric modeling, nor are they adequate to demonstrate the complementarity and necessity of meteorological and image information within the fusion framework. In response to this important comment, we have added a new Section 3.5.3, entitled "Ablation Experiment on Multi-Source Input Data," in the revised manuscript. Under strictly consistent settings of dataset, thickness calibration, and evaluation metrics, we systematically constructed and compared three thickness estimation modes: an image-only mode based solely on geometric information, a meteo-only mode relying exclusively on meteorological variables, and the proposed fusion mode (DTL-IceNet) integrating both image and meteorological information. The experimental results show that neither the image-only nor the meteo-only mode can achieve high-accuracy thickness estimation, whereas the fusion mode significantly outperforms both single-modality configurations in terms of MAPE, PCC, and MSE. These results directly verify the clear complementarity between image geometric information and meteorological process information in icing thickness estimation and confirm that both are indispensable sources of information. Moreover, this ablation study quantitatively distinguishes the independent contributions of the front-end geometric pathway and the back-end meteorological correction pathway within the overall framework, thereby avoiding the misleading conclusion that the final performance is dominated solely by the back-end correction module.

Specific modifications are as follows: **(Page 22-23)**

**3.5.3 Ablation Experiment on Multi-Source Input Data**

**3.5.3 Ablation Experiment on Multi-Source Input Data**

To systematically assess the independent contributions and complementary relationship between image geometric information and meteorological factors in ice thickness estimation, this study conducted an ablation experiment based on the observation field thickness test data. Under the premise of maintaining consistent datasets, thickness calibration methods, and evaluation metrics, three different information configurations were constructed. The first configuration retained only the thickness estimation derived from the segmentation results and area ratio geometric relationship, aiming to measure the independent capability of the image geometric path; the second configuration relied solely on meteorological features such as temperature, humidity, wind speed, and precipitation to directly fit ice thickness using a regression model, evaluating the prediction potential of environmental driving factors without image data; the third configuration introduced the meteorological correction term based on geometric thickness, which is the complete fusion mode of DTL-IceNet proposed in this paper, used to test the practical benefits of the synergistic effects between the two information sources. The comparison results of the three configurations are shown in Table 6.

**Table 6.** Comparison of ablation experimental results based on source input data

| Model Configuration | Image geometry | meteorological elements | MAPE(%) | PCC | MSE |
|---|---|---|---|---|---|
| Image-only | Yes | No | 58.51 | 0.96 | 69.70 |
| Meteo-only | No | Yes | 39.88 | 0.90 | 30.26 |
| **DTL-IceNet(Ours)** | **Yes** | **Yes** | **13.16** | **0.98** | **2.54** |

As seen in Table 6, the three input configurations exhibit clear hierarchical differences in thickness estimation accu- racy, reflecting the complementary nature of image geometry and meteorological factors in terms of information structure. In the Image-only mode, the PCC reaches 0.96, indicating that the geometric thickness derived from the segmentation results and area ratio relationship can well capture the trend of ice variation over time. However, due to the scale uncertainty introduced by single-view imaging, the amplification effect of segmentation errors on area esti- mation, and the simplification of cross-sectional morphology, the magnitude deviation is still significant, with MAPE

reaching 58.51% and MSE reaching 69.70, which reflects the inherent limitations of the geometric path in the absence of environmental process constraints. In the Meteo-only mode, the thickness estimation, relying on the phase changes of meteorological conditions, partially captures the growth and melting rhythm of the ice layer, so the PCC remains at a reasonable level of 0.90. However, due to the lack of spatial volume information, this mode struggles to differen- tiate between absolute thickness differences, exhibiting characteristics of large magnitude errors and significant fluc- tuations. These results indicate that meteorological factors alone cannot provide precise thickness information, espe- cially in scenarios with small-scale changes and significant spatial heterogeneity.

In contrast, DTL-IceNet uses geometric thickness as a spatial scale constraint and employs meteorological features to fit the systemic offset driven by environmental factors, significantly suppressing errors in both trend and magnitude.

The fusion mode's MAPE decreases significantly to 13.16%, MSE reduces to 2.54, and PCC increases to 0.98. This demonstrates that the structured information provided by the geometric path and the phased features captured by the meteorological path are highly complementary in mechanism, with the former determining the spatial baseline for estimation and the latter correcting the deviations caused by changes in meteorological conditions. As the ice for- mation process involves both geometric morphological evolution and meteorological-driven characteristics, both types of information are indispensable. Therefore, a single modality struggles to achieve high accuracy in thickness predic- tion, while the fusion mode can fully leverage the advantages of both types of information, reflecting a dual enhance- ment in robustness and physical consistency.

Comments 6- Minor comments:

Line 141 and Fig9 "glaz"->"glaze"; Table 1 Consider rephrase the table title; Fig13 Consider rephrase the figure title.

**Response:** We sincerely thank the reviewer for the careful and detailed review of our manuscript. In response to the issues you pointed out, we have made the following revisions: all occurrences of "glaz" in the manuscript have been consistently corrected to "glaze"; the title of Table 1 has been revised to "Comparison of icing type recognition performance of various recognition models"; and the title of Figure 13 has been refined with a detailed description for each subfigure. In addition, we have carefully reviewed the entire manuscript and corrected other minor issues accordingly. For brevity, these detailed modifications are not listed one by one in this response.

Specific modifications are as follows:

**Table 1 (Page 17)**

**Table 1.** Comparison of icing type recognition performance of various recognition models

| Modules | Accuracy(%) | W-Prec(%) | W-Recall(%) | W-F1 | M-Prec(%) | M-Recall(%) | M-F1 |
|---|---|---|---|---|---|---|---|
| EfficientNet-V2 | 86.87 | 87.65 | 86.87 | 86.57 | 84.71 | 84.41 | 82.57 |
| MobileNet-V3 | 90.93 | 90.86 | 90.93 | 89.67 | 87.13 | 85.55 | 84.42 |
| ResNeXt | 93.02 | 93.83 | 93.02 | 92.80 | 90.10 | 90.69 | 90.05 |
| MobileOne | 93.41 | 94.26 | 93.41 | 93.21 | 91.21 | 90.71 | 89.27 |
| **ResSepNet(Ours)** | **95.23** | **96.44** | **95.23** | **94.84** | **92.84** | **91.84** | **90.84** |

**Figure 13 (Page 18)**

[Figure]

**Figure 13.** Comparison of segmentation performance across different models. (a) Segmentation result under daytime conditions (first image), (b) Segmentation result under daytime conditions (second image), (c) Segmentation result under nighttime conditions (first image), and (d) Segmentation result under nighttime conditions (second image).

---

## Author Comment (AC2)

Thank you very much for taking the time to review this manuscript. I have responded to each of the issues raised by the two reviewers and have revised the manuscript.

**Response to Reviewer 2 Comments**

Comments 1- Please provide detailed information for different branch in ResSepNet in Section 3.3.1. Has results in Figure 9 been statistically tested? Please include more information on this.

**Response:** We sincerely thank the reviewer for the valuable comments regarding the structural details and experimental rigor of ResSepNet. In response to your concerns, we have added a detailed description of the design principles and functional roles of each branch in Section 3.3.1, "ResSepNet Branch Ablation Experiment" in the revised manuscript. Specifically, the background branch is designed to extract environmental features and suppress background noise, the icing branch focuses on capturing detailed and textural features of the icing regions on transmission lines, and the global branch employs EfficientNet-B3 to extract global contextual information from the entire image. These multi-scale features are then normalized and fused to enhance the discriminative capability for ice type recognition. In addition, to address the statistical reliability of the results shown in Figure 9, we clarified that the reported results are the average of five independent experimental runs conducted under identical training and testing datasets and hardware environments, ensuring good comparability and consistency. The small variation among repeated experiments further demonstrates the high stability of the proposed model.

Specific modifications are as follows: **(Page 14; Page 15)**

**3.3.1 ResSepNet Branch Ablation Experiment**

**3.3.1 ResSepNet Branch Ablation Experiment**

In ResSepNet, the ice type recognition task is achieved through the collaborative efforts of the background branch,
icing branch, and global branch. The background branch preprocesses the original image to extract the background
subgraph, focusing on capturing environmental features in the image while minimizing the interference of background
noise. The icing branch, on the other hand, specializes in extracting features of the icing on the transmission line. It
uses a structure similar to that of the background branch but places greater emphasis on capturing the details and
texture information of the icing area. The global branch directly inputs the entire image, utilizing EfficientNet-B3 as
the backbone network to extract macro features from the full image and capture global context information through a
transfer learning model. The design of these three branches aims to capture both local and global information at dif-
ferent spatial scales by performing feature extraction in different regions, thereby effectively reducing the influence
of background noise and improving the accuracy of ice type recognition. By normalizing and fusing the features
extracted from the different branches, ResSepNet can fully leverage the spatial scale information extracted by each
branch, ultimately leading to more accurate ice type recognition results.

**Figure 9.** Confusion matrix of ice type recognition effect of each branch of ResSepNet.

For the results in Figure 9, five independent experimental runs were conducted and the average values were reported
to ensure data stability and reliability. All runs used the same training and test datasets and were performed under
identical hardware conditions to guarantee consistency. Thus, the confusion matrix in Figure 9 represents the averaged
results with minimal variation, indicating high model stability across repeated experiments. Based on the confusion

Comments 2- The paper currently lacks a dedicated "Discussion" section, which limits the depth and completeness of the research. It is recommended that the authors add this section, focusing on the following three key aspects: Model Advantages, Mechanism Interpretation and Model Limitations.

**Response:** We sincerely thank the reviewer for pointing out the absence of a dedicated Discussion section in the original manuscript. Following your suggestion, we have newly added and systematically constructed Section 4 Discussion in the revised version, where the proposed method is analyzed in depth from three key aspects: model advantages, mechanism interpretation, and model limitations. Regarding model advantages, we discuss the overall strengths of DTL-IceNet in ice type recognition, ice region segmentation, and ice thickness estimation from the perspectives of multi-task collaboration and multi-source information fusion, highlighting its improvements over existing methods. With respect to mechanism interpretation, we analyze the intrinsic reasons for the strong performance of the model along three main lines, namely multi-branch feature extraction, multi-scale attention mechanisms, and the collaborative modeling of geometric and meteorological constraints. In terms of model limitations, we further discuss the fact that the current dataset does not yet cover extreme weather or complex terrain conditions, that thickness validation is only based on clear-sky observation-site data, and that the dual-view structure has not been fully validated under real side-view imaging conditions, and we objectively analyze their potential impacts on model generalization. In addition, we combine different imaging conditions and meteorological factors to investigate, at the mechanism level, the possible error sources of the model in complex environments. The Discussion section also explicitly outlines future research directions involving extreme weather scenarios, multi-terrain environments, and real dual-view experimental platforms. Overall, the newly added Discussion provides a comprehensive and critical reflection on this work from the perspectives of advantages, mechanisms, limitations, and future developments.

Specific modifications are as follows: **(Page 23-25)**

**4 Discussion**

**4 Discussion**

The dual-task learning framework, DTL-IceNet, proposed in this study demonstrates high reliability in both ice type recognition and thickness detection tasks. Its main contribution lies in the unified modeling of three types of information: type, geometry, and meteorology, which effectively enhances the comprehensive sensing ability of icing conditions on transmission lines. This fusion-based design aligns with the view emphasized in the literature that "ice physics, image features, and environmental processes must be considered in a coordinated manner" (Fan et al., 2018; Hao et al., 2023; Chen et al., 2024; Dong et al., 2022; Meng et al., 2025; Han et al., 2024), and it achieves both discriminative ability and physical consistency in typical monitoring scenarios, leading to significant improvements over existing methods.

In the ice type recognition task, the model explicitly separates the background, conductor, and icing regions through a multi-branch feature extraction structure. Studies by Fan et al. (2018). and Hao et al. (2023) have pointed out that ice recognition is highly sensitive to the environmental background, and deep networks based on a single-path feature extraction often struggle to fully capture the local texture of the conductor in complex backgrounds. The decomposition-based modeling approach of DTL-IceNet significantly enhances the distinction between different types of icing under complex lighting, fog, and noise conditions, providing stronger anti-interference ability compared to single-branch methods.

In the ice region segmentation task, MOMSA-SegNet leverages a multi-scale attention module to improve the representation capability of thin conductors and irregularly shaped icing areas. Existing research has shown that traditional edge detection or low-level feature methods exhibit poor robustness in weak boundaries, low contrast, and nighttime scenarios (Han et al., 2024; Tan & Le, 2019; Hu et al., 2018; Vasu et al., 2023; Li et al., 2016). In contrast, the multi-scale attention mechanism effectively utilizes the contextual structure surrounding the transmission line, allowing the model to maintain stable geometric contour predictions in typical scenes such as sunny, foggy, and nighttime conditions. Its accuracy advantage stems from the targeted utilization of the transmission line image structure rather than relying solely on the depth of the network or the scale of parameters.

For ice thickness estimation, the improvement in model performance is mainly attributed to the synergistic effect of geometric and meteorological constraints. The image geometric information provides the basic trend of ice volume changes, but relying solely on 2D images cannot accurately reflect the true 3D shape of the ice, leading to systematic biases during temperature, humidity, wind speed, and precipitation phases. By introducing the meteorological correction term based on environmental factors, the model performs consistent corrections on the initial thickness estimation according to the basic physical laws of ice growth and melting, effectively compensating for the inherent structural biases in the geometric estimation. Experimental results show that the thickness curves align more closely with the actual distribution across multiple phases, indicating the complementary role of geometric and meteorological information in thickness estimation.

Although DTL-IceNet demonstrates robustness under typical monitoring conditions, its applicability is still limited by the data coverage and experimental conditions. The data used in this study were primarily collected in typical meteorological scenarios and have not yet covered extreme weather conditions such as severe convection or freezing rain coupled with strong winds. Under such conditions, the signal-to-noise ratio of images and the rate of ice morphology change may exceed the training distribution, and the model's robustness needs further validation. Moreover, the current experimental data do not systematically reflect complex spatial environments with significant topographic variations. Existing studies have shown that terrain, especially in valley wind fields, significantly impacts icing distribution, so the model's performance in complex terrain scenarios remains uncertain.

While the segmentation experiments presented in this study systematically show performance differences under
various imaging conditions such as sunny days, fog, and nighttime, the thickness validation experiments were limited
by the actual conditions of the observation field, which only included real thickness calibration data from sunny sce-
narios and could not directly quantify thickness estimation accuracy under various weather conditions. To address this
logical gap, we will further analyze the potential impact of weather factors on thickness estimation. Mechanistically,
weather changes affect thickness results mainly through two paths. First, imaging degradation such as fog and low
light reduces the clarity of the ice and conductor boundaries, causing segmentation masks to deviate in geometric
details and directly affecting the preliminary thickness calculation based on area and contour inference. Second, me-
teorological variables such as temperature, humidity, and wind speed determine the growth and melting rates of the
ice, strongly driving the temporal evolution of thickness. The meteorological correction module of DTL-IceNet can
provide systematic corrections based on this driving pattern, but it cannot completely compensate for geometric devi-
ations caused by severe imaging degradation. Therefore, under real fog or nighttime conditions, the initial thickness
errors may increase, while the optimized thickness is expected to show more stable but still limited corrections.

The dual-perspective approach proposed in this study is one of the key innovations of the overall framework.
Through joint segmentation of the main and side perspectives, the major and minor axes of the ice-covered cross-
section can be theoretically estimated, improving the certainty of geometric parameters. However, it is important to
note that the thickness validation experiment in this paper was limited by site constraints and did not actually deploy
transmission lines for real side-perspective imaging. Thickness detection relied on a simplified geometric assumption
primarily based on the main perspective. This experimental condition means that the final obtained thickness accuracy
mainly reflects the performance under the simplified single-perspective system rather than the upper limit of a com-
plete dual-perspective structure. Therefore, while the current results prove the feasibility and potential value of the
proposed framework, they do not fully validate the information gain of the dual-perspective structure in real multi-
conductor scenarios. This also suggests that the side-perspective segmentation error's impact on thickness calculation
has not been fully quantified. Future work will focus on building an experimental platform that truly reflects the dual-
perspective structure to systematically evaluate error propagation mechanisms and further optimize geometric fusion
methods.

In summary, DTL-IceNet forms a cohesive solution across ice type recognition, geometric structure extraction, and
thickness estimation, with its advantages derived from complementary constraints between tasks and explicit incor-
poration of physical processes.

**Comments 3-** The connection between the high mIoU of MOMSA-SegNet and the final thickness MAPE is not sufficiently demonstrated. To validate the overall framework, it is essential to provide evidence that the superior segmentation performance is a prerequisite for the low thickness error. Ablation studies comparing the impact of different segmentation qualities on the final MAPE are needed.

**Response:** We sincerely thank the reviewer for the professional and highly targeted suggestions. In response to your comments, we have added a new Section 3.5.2, entitled "Effect of Different Segmentation Models on Ice Thickness Detection," in the revised manuscript. Under strictly identical ice-type recognition and thickness optimization procedures, UNet++, SegNet, DySample, and the proposed MOMSA-SegNet are respectively employed as the front-end segmentation module, and comparative thickness estimation experiments are conducted on the same observation-site test dataset. The impacts of different segmentation qualities on both the initial and optimized thickness errors are systematically evaluated. The experimental results (Table 5) demonstrate that better segmentation performance consistently leads to lower MAPE and MSE for both the initial and optimized thickness estimates, and the thickness estimation based on MOMSA-SegNet achieves the best performance across all evaluation metrics. Moreover, even after the introduction of meteorological correction, the error differences among different segmentation models remain evident, indicating that the geometric structural errors introduced at the segmentation stage cannot be fully eliminated by the back-end meteorological correction. These results directly verify, under unified experimental conditions, that high segmentation accuracy is a necessary prerequisite for achieving low thickness estimation error, and quantitatively reveal the decisive influence of segmentation quality on the final thickness estimation performance.

Specific modifications are as follows: **(Page 21-22)**

**3.5.2 Effect of Different Segmentation Models on Ice Thickness Detection**

**3.5.2 Effect of Different Segmentation Models on Ice Thickness Detection**

To quantitatively evaluate the impact of the segmentation module on the accuracy of final ice thickness estimation, this study, under the premise of maintaining consistency in the ice type recognition and thickness optimization calcu- lation processes, uses UNet++, SegNet, DySample, and MOMSA-SegNet as the frontend segmentation models. The thickness estimation experiments were conducted on the same observation field test dataset. The ice type recognition module and thickness optimization calculation module were kept unchanged, and only the ice segmentation submod- ule was replaced with different typical methods. The initial thickness and optimized thickness errors were then com- puted on the corresponding observation field thickness test data. This approach allows a direct comparison of the effects of different segmentation models on geometric scale calculation and error propagation within a unified frame- work, providing a clearer insight into the structural relationship between segmentation quality and thickness detection accuracy. The experimental results are shown in Table 5.

**Table 5.** Comparison of ice thickness detection performance driven by different segmentation models

| Modules | Initial ice thickness MAPE(%) | Optimized ice thickness MAPE(%) | Initial ice thickness MSE | Optimized ice thickness MSE |
|---|---|---|---|---|
| UNet++ | 69.53 | 14.58 | 80.93 | 2.26 |
| SegNet | 65.38 | 13.71 | 76.09 | 2.12 |
| DySample | 57.45 | 12.05 | 66.86 | 1.87 |
| **MOMSA-SegNet(Ours)** | **56.36** | **11.82** | **65.60** | **1.83** |

As seen in Table 5, under the same ice type recognition and meteorological correction processes, the performance of ice thickness detection shows a consistent trend with different segmentation models as frontend submodules. The better the segmentation performance, the lower the initial and optimized thickness errors. When MOMSA-SegNet, the segmentation model proposed in this paper, is used, both the MAPE and MSE of the initial thickness are the lowest.

After replacing it with UNet++, SegNet, or DySample, both errors increase to varying degrees. This indicates that the segmentation stage directly affects the accuracy of the ice contour and area depiction, which in turn influences the downstream geometric parameter estimation. Segmentation errors accumulate and amplify in the calculation of equiv- alent ice thickness.

A further comparison of the initial and optimized thickness metrics reveals that after the introduction of meteorological correction, the MAPE and MSE for all models significantly decrease, showing that the environmental-driven correction terms can effectively compensate for the system errors caused by the 2D perspective and geometric simplifications. However, the relative differences between the models still remain even after optimization. Even with meteorological correction, the thickness estimation based on MOMSA-SegNet maintains the lowest MAPE and MSE, while segmentation models with weaker performance still exhibit higher optimization errors. This suggests that the meteorological correction mainly targets global system biases related to environmental processes, and cannot fully counteract the structural errors in the ice region contour and scale caused by segmentation. Thus, it can be concluded that segmentation quality determines the geometric baseline for thickness estimation, while meteorological correction fine-tunes this baseline, forming a hierarchical complementary relationship.

In conclusion, under the condition of complete consistency in the thickness estimation algorithm and test dataset, simply replacing the segmentation module results in a monotonous or nearly monotonous decrease in thickness errors with the improvement of segmentation model performance. This result confirms the substantial contribution of MOMSA-SegNet in ice thickness detection from a data-driven perspective. Its higher segmentation accuracy not only reflects in pixel-level metrics but also significantly reduces the initial errors in downstream thickness estimation, maintaining its advantage even after meteorological correction and effectively transmitting the improved segmentation performance to the final physical quantity estimation results.